# Associations between birth weight and adult apolipoproteins: The LifeGene cohort

**Shantanu Sharma**[1]*, **Louise Bennet**[1,2], **Agne Laucyte-Cibulskiene**[1,3,4], **Anders Christensson**[1,3], **Peter M. Nilsson**[1,5]

**1** Department of Clinical Sciences in Malmö, Lund University, Lund, Sweden, **2** Clinical Trials Unit, Skåne University Hospital, Lund, Sweden, **3** Department of Nephrology, Skåne University Hospital, Malmö, Sweden, **4** Department of Clinical Science, Intervention and Technology, Division of Renal Medicine, Karolinska Institute, Stockholm, Sweden, **5** Department of Internal Medicine, Research Unit, Skåne University Hospital, Malmö, Sweden

* shantanu.sharma@med.lu.se

## Abstract

### Background

Early life factors may predict cardiovascular disease (CVD), but the pathways are still unclear. There is emerging evidence of an association of early life factors with apolipoproteins, which are linked to CVD. The study objective was to assess the associations between birth variables and adult apolipoproteins (apoA1 and apoB, and their ratio) in a population-based cohort.

### Methods

The LifeGene Study is a prospective cohort comprising index participants randomly sampled from the general population. Blood samples were collected between 2009 and 2016. In this sub-study, we used birth variables, obtained from a national registry for all participants born 1973 or later, including birth weight and gestational age, while adult CVD risk factors included age, sex, body mass index (BMI), lipids, and smoking history. We employed univariate and multivariate general linear regression to explore associations between birth variables, lipid levels and other adult CVD risk factors. The outcomes included non-fasting apoA1 and apoB and their ratio, as well as total cholesterol and triglycerides. A total of 10,093 participants with both birth information and lipoprotein levels at screening were included. Of these, nearly 42.5% were men (n = 4292) and 57.5% were women (n = 5801).

### Results

The mean (standard deviation) age of men was 30.2 (5.7) years, and for women 28.9 (5.8) years. There was an increase of 0.022 g/L in apoA1 levels per 1 kg increase in birth weight (p = 0.005) after adjusting for age, sex, BMI, gestational age, and smoking history. Similarly, there was a decrease of 0.023 g/L in apoB levels per 1 kg increase in birth weight (p<0.001) after adjusting for the same variables. There were inverse associations of birth weight with the apoB/apoA1 ratio. No independent association was found with total cholesterol, but with

**Data Availability Statement:** The anonymized minimal dataset is available. For details about the LifeGene cohort and instructions on how to apply

for data, see link: https://lifegene.se/for-scientists/apply-for-data/.

**Funding:** The LifeGene Study has been funded by the Torsten and Ragnar Söderbergs Foundation, AFA Insurance, Karolinska Institute, the Stockholm County Council, and the Swedish Research Council. The funders had no role in study design, data collection and analysis, decision to publish, or preparation of the manuscript.

**Competing interests:** The authors have declared that no competing interests exist.

triglyceride levels (□-coefficient (95% Confidence Interval); -0.067 (-0.114, -0.021); p-value 0.005).

## Conclusions

Lower birth weight was associated with an adverse adult apolipoprotein pattern, i.e., a higher apoB/apoA1 ratio, indicating increased risk of future CVD manifestations. The study highlights the need of preconception care and pregnancy interventions that aim at improving maternal and child outcomes with long-term impacts for prevention of cardiovascular disease by influencing lipid levels.

## Introduction

Apolipoproteins constitute a group of multifunctional proteins that play a vital role in lipid homeostasis [1, 2]. Among the different clinically important apolipoproteins linked to lipid metabolism, apoB and apoA1 are crucial components of lipoproteins such as very low-density lipoproteins (VLDL), low-density lipoproteins (LDL), and high-density lipoproteins (HDL). ApoB levels are highly correlated with LDL-cholesterol and non-HDL cholesterol levels [2]. Furthermore, apoB has been found to be associated with the onset of adult cardiovascular diseases (CVD) caused by atherosclerosis and metabolic disturbances such as diabetes, hyperinsulinemia, etc. [3]. On the contrary, apoA1 functions as a major structural component of HDL-cholesterol, as one component of the metabolic syndrome (dyslipidemia, hypertension, and abdominal obesity). Empirical evidence suggests a strong inverse observational relationship between HDL-cholesterol and cardiovascular risk [2, 4]. McQueen *et al*. highlighted in their study that the apoB to apoA1 ratio is a better risk marker of coronary artery disease compared to apoB or apoA1 alone [4]. An imbalance between apoB and apoA1 resulting in an increased apoB/apoA1 ratio was also strongly associated with CVD risk in a recent large study (n = 137 000) from Sweden in both men and women of all ages [5].

Multiple factors, including lifestyle, dietary habits, and genetic makeup, influence the serum concentration of apolipoproteins [6, 7]. Furthermore, there is emerging evidence of early life influences on apolipoprotein concentrations in adulthood [8, 9]. According to the Developmental Origins of Health and Disease (DOHaD) hypothesis, adverse intrauterine conditions contribute to the programming of future health and diseases in fetuses, including their effect on the liver function that affects lipid and glucose metabolism [10].

Considering the limited and inconclusive evidence for the effect of birth variables on adult apolipoproteins, we conducted an analysis on data obtained from a Swedish sample of young and middle-aged adults. The objective and primary aim of our study was to assess the associations between birth variables (birth weight, and with adjustments for gestational age and other confounders) and adult apolipoproteins (apoA1, apoB, and their ratio) as well as conventional lipid variables in a sub-study of the LifeGene cohort.

## Materials and methods

### Study participants

The LifeGene Study is a population-based prospective cohort. It consists of index people aged 18 years or older randomly sampled from the general population as well as individuals who spontaneously registered for participation [11]. Data were collected through a comprehensive

web-based questionnaire comprising multifaceted questions concerning phenotypes and exposures as well as physical measurements and blood sample collected at test centers. Index participants were encouraged to invite their household members, e.g., their partners or children. Informed written consent was obtained from participants, either electronically or on site when providing the blood samples. The participants could book an appointment at the test center of the LifeGene study for testing on site as a supplement to the online questionnaire. The data were collected between 2009 and 2016 and stored in a central database at Karolinska Institute, Stockholm, Sweden. LifeGene examination procedures are described in https://lifegene.se/wp-content/uploads/1LifeGeneresource20170203version24.pdf.

The web-based questionnaire had nine themes, including socio-demography, lifestyle, self-care, women's health, living habits, health history, asthma and allergies, injuries, and mental health. The physical measurements included body weight (kg) and height (m), waist, hip, and chest circumference (cm), heart rate (beats/min), and blood pressure (mmHg). Lithium heparin tubes were obtained for analysis of a battery of clinical chemistry non-fasting parameters, including apoA1 and apoB, as well as total cholesterol and triglycerides. The blood samples were collected in a non-fasting state.

In this sub-study, a sample of LifeGene participants born in 1973 or later was linked (using the 10-digit personal identification number assigned to all born in Sweden) to the Swedish National Medical Birth Register, which has information on the birth characteristics of both the mother and the child for all births in Sweden since 1973 [12]. For this study, we focused on birth weight and gestational age for all LifeGene participants 18 years or older at the time of testing in LifeGene. A flowchart of the selection of the sample is presented in **Fig 1.** A total of 10,121 participants with apolipoprotein analyses were included, and after linkage to the National Medical Birth Register, 10,093 participants remained for the current analyses. The samples of finally included participants (n = 10093) were collected between 18/04/2011 and 22/12/2016 and analyzed in the laboratories immediately. The statistical analysis of the data was performed between 2022–2023. We did not have access to information that could identify individual participants during or after data collection.

## Study variables

The main dependent primary variables included apoA1, apoB, and their calculated ratio, as well as total cholesterol and triglycerides. The independent variables included *birth variables* (birth weight and gestational age) and *adult CVD risk factors* such as adult age, sex, and body mass index (BMI). We also asked about the smoking history ("*Have you smoked more than 100 cigarettes in your entire life*?"); the question has been used in some of the previous surveys [13, 14].

## Study procedures

The non-fasting blood samples were drawn at LifeGene test centers. The analyses of blood samples for lipids (total cholesterol, triglyceride, apolipoproteins) were done by use of standard methods (immunochemistry, turbidimetry), firstly at Unilabs, St. Göran Hospital, Stockholm, between 2009 and March 2010, and secondly at Dept. Clinical Chemistry, Karolinska University Hospital, between 2010 and 2016. However, the triglycerides levels were only collected for those tested between October 2009 and March 2010. No LDL- or HDL cholesterol tests were carried out as the apolipoproteins replaced these variables.

## Definitions

The BMI was calculated using the formula: weight in kg/height in meters$^2$.

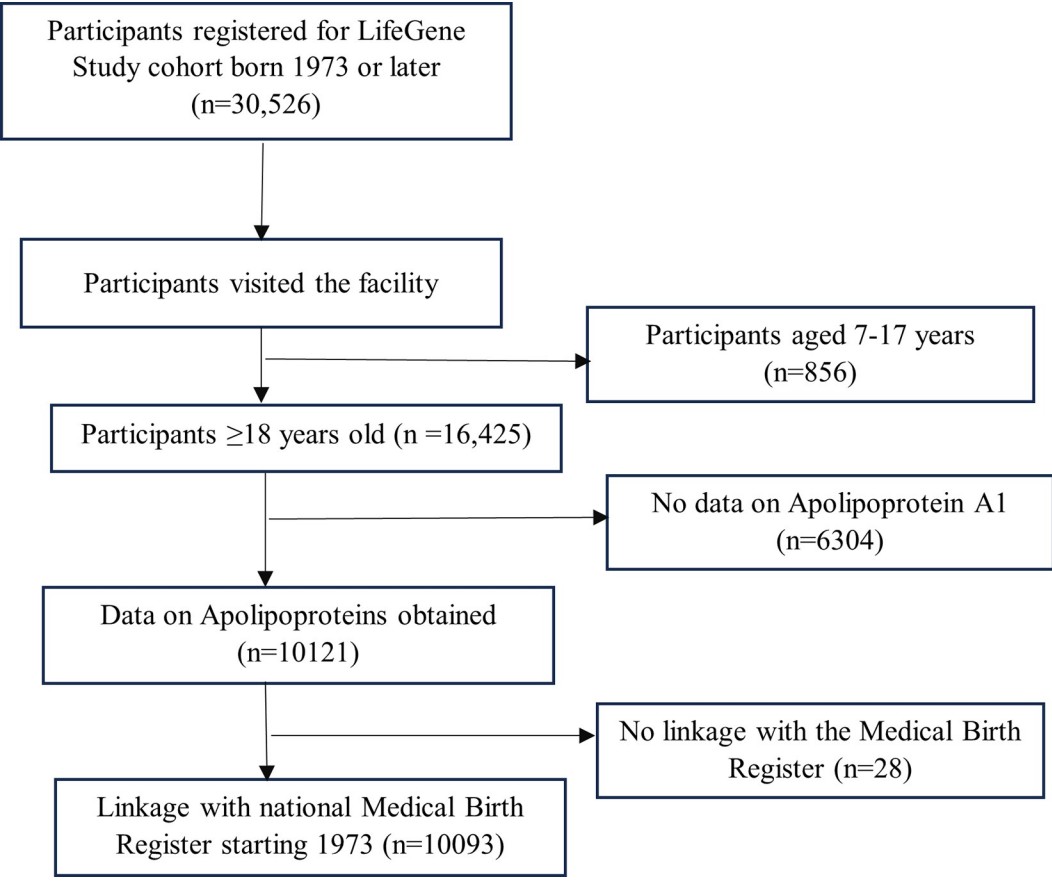

**Fig 1. Flow chart of the study sample from the LifeGene study cohort (n = 10093).**

ApoB/apoA1 ratio was estimated by dividing apoB levels by apoA1.

## Inclusion criteria

All index persons 18 years or older signing informed consent had blood sampling and if born in 1973 or thereafter, irrespective of sex or ethnicity, were included in the present analysis.

## Statistical analysis

The numerical descriptive data were presented as means with standard deviation, SD (if normally distributed), or median with interquartile range (if skewed distribution). The categorical descriptive data were presented as frequencies and percentages. All the variables were normally distributed except for triglyceride levels, which were log-transformed for use in regression analysis. We employed univariate or multivariate general linear regression to explore associations between the dependent and independent variables. Full factorial models were run in the general linear regression analysis. The strength and direction of the association were expressed as β coefficients with a 95% Confidence Interval (95% CI).

In the multivariate general linear regression, stepwise models were analyzed. *Model I* was adjusted for gestational age and sex. *Model II*, BMI, and adult age were added to the variables adjusted in Model I. Similarly, *Model III*, smoking history was added to the variables adjusted for in Model II. All the statistical analyses were performed in Statistical Package for the Social

Sciences (IBM SPSS Statistics for Windows, Version 27.0. Armonk, NY: IBM Corp). A p-value <0.05 was considered statistically significant.

## Ethical approval

This study was approved by the Ethics Review Authority (Etikprövningsmyndigheten), Sweden (Dnr: 2019–02408), based on an earlier approval of the LifeGene study from the Ethics Review Board at Karolinska Institute (Dnr: 2009/615-31/1), and in addition the LifeGene legal permission by Swedish law (Lag 2013:794).

## Results

A total of 10,093 participants who had birth information and apolipoprotein levels were included in the study. The majority (57.5%) of the participants were women (n = 5801). The mean (SD) age of the men was 30.25 (5.73) years, and for women, 28.95 (5.84) years, as shown in **Table 1**. The mean apoA1 levels among women were 1.68 g/L, and in men, 1.46 g/L, and the mean apoB levels among men were 0.86 g/L and in women 0.78 g/L. There was an inverse relationship between birth weight and apoA1 levels (β-coefficient (95%CI) -0.018 [-0.029, -0.007]; p = 0.001) in the unadjusted model; however, the association turned positive after adjusting for all the covariates and confounders, **Table 2**. Hence, there was an increase of 0.022 g/L in apoA1 levels per 1 kg increase in birth weight (*Model III*).

For apoB, there was an inverse association with birth weight (β-coefficient (95%CI) -0.021 [-0.030, -0.012]; p<0.001) after adjusting for gestational age and sex (**Table 3**). Hence, there was a decrease of 0.023 g/L in apoB levels per 1 kg increase in birth weight after full adjustment (*Model III*).

Likewise, there was an inverse association of birth weight with the apoB/apoA1 ratio (β-coefficient (95%CI) -0.024 [-0.032, -0.015]; p<0.001) after full adjustments for cofounders and covariates (**Table 4**).

No statistically significant association of birth weight was noticed with total cholesterol levels (**Table 5**). However, there were inverse associations of birth weight with triglyceride levels (β-coefficient (95%CI) -0.067 [-0.114, -0.021]; p = 0.005) after adjustments for all the cofounders and covariates (**Table 6**).

## Discussion

In this population-based study of young and middle-aged adults, we assessed the effect of birth weight (unadjusted or adjusted for gestational age and other covariates or confounders) on adult lipoproteins, apoA1 and apoB levels, and their ratio, but also on (non-fasting) total cholesterol and triglycerides. We found an inverse association between birth weight and apoB and a positive association with apoA1 levels after full adjustment. This could be interpreted that adults born with lower birth weight relative to gestational age may have higher apoB and lower apoA1 levels in adulthood, a documented risk pattern for CVD in the Swedish population [5]. This study is one of the few that explored such associations [15–17], and the largest so far. Barker *et al.* also reported higher apoB levels in children born with low birth weight [16] and highlighted that impaired growth of the liver in late gestation leads to permanent changes in the low-density lipoprotein cholesterol metabolism [16]. Furthermore, Barker *et al.* showed that with a 1 mm increase in abdominal circumference at birth, there was a 0.04 g/L decrease in serum apoB concentration in adults aged 50–53 years; however, the study sample included only 219 men and women [16]. In categories of birth weight, higher levels of apoB were associated with lower birth weight [16]. On the other hand, Starnberg *et al.*, in their study among

**Table 1. Descriptive data of men and women with birth weight data in the LifeGene cohort.**

| Variables | Men (n = 4292) mean (SD) | Women (n = 5801) mean (SD) |
|---|---|---|
| **Age (years)** | 30.25 (5.73) | 28.95 (5.84) |
| Missing | 6 | 129 |
| **Birth weight (kg)** | 3.59 (0.53) | 3.45 (0.50) |
| Missing | 0 | 0 |
| **Gestational age (weeks)** | 39.56 (1.80) | 39.58 (1.71) |
| Missing | 19 | 20 |
| **Adult BMI (kg/m$^2$)** | 24.45 (2.98) | 22.73 (3.21) |
| Missing | 6 | 129 |
| **SBP (mmHg)** | 120.66 (10.37) | 110.15 (9.55) |
| Missing | 1 | 2 |
| **Total cholesterol (mmol/L)** | 4.71 (0.94) | 4.62 (0.81) |
| Missing | 0 | 0 |
| **Triglycerides levels (mmol/L)** | 1.10 (0.77–1.60) | 0.78 (0.58–1.10) |
| N (sample)* | 1373 | 2013 |
| **apoA1 (g/L)** | 1.46 (0.22) | 1.68 (0.29) |
| Missing | 0 | 0 |
| **apoB (g/L)** | 0.86 (0.22) | 0.78 (0.18) |
| Missing | 12 | 23 |
| **apoB/apoA1 ratio** | 0.61 (0.19) | 0.48 (0.13) |
| Missing | 12 | 23 |
| **Have you smoked more than 100** | | |
| **cigarettes in your entire life?$** | | |
| • Yes | 1554 (36.20) | 2122 (36.58) |
| • No | 1082 (25.21) | 1333 (22.97) |
| • Do not know/refuse | 23 (0.53) | 46 (0.79) |
| • Missing | 1633 (38.04) | 2300 (39.64) |

*The number of participants with triglyceride levels was limited and did not always include the same individuals as those with apolipoprotein levels. $The data were presented as N(%)*

Continuous normally distributed variables presented as Mean (Standard Deviation), non-normally distributed–as Median (Interquartile Range).

*Abbreviations*: apoA1: Apolipoprotein A1; apoB: Apolipoprotein B; BMI: Body Mass Index; SBP: Systolic Blood Pressure; SD: Standard Deviation

Swedish children of 7 years of age, did not show any significant association between low birth weight and apoB or apoA1 levels [17].

As a previous large meta-analysis could not show any significant association between birth weight and total cholesterol [18], even if contradicted by a more recent but smaller meta-analysis in adults only that found an inverse association [19], it is of importance to use more detailed information on lipid patterns. For example, the apolipoproteins in larger cohorts such as ours, to elucidate on early life influences on lipids. The meta-analysis by Huxley *et al.* reported varying directions of associations between birth weight and adult total cholesterol but included just a few smaller and inconclusive studies on apolipoproteins [18]. Almost all the studies in the meta-analysis reported apoA1 and apoB findings in children or adolescents but not in adults >20 years, except for the study by Roseboom *et al.* This Dutch famine study found a statistically significant association of birth weight and ponderal index of individuals born during the famine with apoA1 levels but no significant association with apoB levels in adulthood [20].

**Table 2. General linear regression model of the association between birth weight (adjusted for confounders and covariates) and *apoA1 levels* (dependent variable).**

| Variables | Unadjusted Model | Model I | Model II | Model III* |
|---|---|---|---|---|
| | β-coeff. (95%CI); p-value | β-coeff. (95%CI); p-value | β-coeff. (95%CI); p-value | β-coeff. (95%CI); p-value |
| **Birth weight** (kg) | **-0.018 (-0.029, -0.007); 0.001** | **0.013 (0.001, 0.025); 0.030** | **0.024 (0.012, 0.036); <0.001** | **0.022 (0.007, 0.036); 0.005** |
| **Gestational age** (weeks) | - | -0.002 (-0.006, 0.001); 0.205 | **-0.004 (-0.008, -0.001); 0.014** | **-0.005 (-0.010, -0.001); 0.022** |
| **Sex** | - | | | |
| • Women | | **0.227 (0.216, 0.238); <0.001** | **0.204 (0.193, 0.215); <0.001** | **0.202 (0.188, 0.216); <0.001** |
| • Men | | Reference | Reference | Reference |
| **BMI** (kg/m$^2$) | - | - | **-0.014 (-0.016, -0.013); <0.001** | **-0.015 (-0.018, -0.013); <0.001** |
| **Age** (years) | - | - | **0.003 (0.002, 0.004); <0.001** | **0.003 (0.002, 0.004); <0.001** |
| **Smoking history** | - | - | - | |
| • Yes | | | | **0.019 (0.005, 0.032); 0.008** |
| • Don't know/refused | | | | 0.028 (-0.036, 0.091); 0.394 |
| • No | | | | Reference |

*Model I*, birth weight was adjusted for gestational age and sex. In *Model II*, apart from gestational age and sex, BMI and adult age was adjusted for. Similarly, in *Model III*, apart from variables adjusted for in *Model I* and *II*, history of smoking was added.

*R$^2$ = 17.2%

*Abbreviations*: apoA1: Apolipoprotein A1; BMI: Body Mass Index; β-coeff: β-coefficient; CI: Confidence Interval

P-value<0.05 was considered statistically significant, with significant associations in bold.

Our results are congruent with other studies. Leger *et al.*, reported that children born small for gestational age had lower mean apoA1 and higher apoB levels in young adulthood (in their twenties) compared to those born appropriate for gestational age [21]. Similarly, Roseboom *et al.*, in their study among people exposed or non-exposed to the Dutch Winter famine in 1944–1945, found a positive association between birth weight and adult apoA1 levels [20]. The

**Table 3. Generalized linear regression model of association between birth weight (adjusted for confounders and covariates) and *apoB levels* (dependent variable).**

| Variables | Unadjusted Model | Model I | Model II | Model III* |
|---|---|---|---|---|
| | β-coeff. (95%CI); p-value | β-coeff. (95%CI); p-value | β-coeff. (95%CI); p-value | β-coeff. (95%CI); p-value |
| **Birth weight** (kg) | -0.004 (-0.012, 0.004); 0.289 | **-0.021 (-0.030, -0.012); <0.001** | **-0.020 (-0.029, -0.011); <0.001** | **-0.023 (-0.034, -0.012); <0.001** |
| **Gestational age** (weeks) | - | **0.004 (0.001, 0.007); 0.003** | 0.002 (-0.001, 0.004); 0.161 | 0.001 (-0.002, 0.004); 0.472 |
| **Sex** | - | | | |
| • Women | | **-0.082 (-0.091, -0.074); <0.001** | **-0.052 (-0.060, -0.044); <0.001** | **-0.055 (-0.065, -0.045); <0.001** |
| - Men | | Reference | Reference | Reference |
| **BMI** (kg/m$^2$) | - | - | **0.013 (0.012, 0.015); <0.001** | **0.014 (0.012, 0.015); <0.001** |
| **Age** (years) | - | - | **0.007 (0.006, 0.007); <0.001** | **0.007 (0.006, 0.008); <0.001** |
| **Smoking history** | - | - | - | |
| • Yes | | | | -0.003 (-0.013, 0.007); 0.517 |
| • Don't | | | | 0.009 (-0.038, 0.056); 0.708 |
| • know/refused | | | | Reference |
| • No | | | | |

*Model I*, birth weight was adjusted for gestational age and sex. In *Model II*, apart from gestational age and sex, BMI and adult age was adjusted for. Similarly, in *Model III*, apart from variables adjusted in *Model I* and *II*, history of smoking was added.

*R$^2$ = 13.2%

*Abbreviations*: apoB: Apolipoprotein B; BMI: Body Mass Index; β-coeff: β-coefficient; CI: Confidence Interval

P-value<0.05 was considered statistically significant, with significant associations in bold.

**Table 4. Unadjusted and adjusted model of linear regression between *apoB/apoA1 ratio* levels (dependent variable) and independent variables.**

| Variables | Unadjusted Model | Model I | Model II | Model III* |
|---|---|---|---|---|
| | β-coeff. (95% CI); p-value | β-coeff. (95% CI); p-value | β-coeff. (95% CI); p-value | β-coeff. (95% CI); p-value |
| **Birth weight** (kg) | 0.002 (-0.004, 0.009); 0.496 | **-0.019 (-0.027, -0.012); <0.001** | **-0.023 (-0.030, -0.016); <0.001** | **-0.024 (-0.032, -0.015); <0.001** |
| **Gestational age** (weeks) | | **0.004 (0.001, 0.006); 0.001** | **0.003 (0.001, 0.005); 0.008** | **0.003 (0.00, 0.006); 0.025** |
| **Sex** | | | | |
| Women | | **-0.132 (-0.139, -0.125); <0.001** | **-0.103 (-0.110, -0.097); <0.001** | **-0.102 (-0.110, -0.094); <0.001** |
| | | | | *Reference* |
| Men | | *Reference* | *Reference* | |
| **BMI** (kg/m$^2$) | | | **0.014 (0.013, 0.015); <0.001** | **0.015 (0.014, 0.016); <0.001** |
| **Age** (years) | | | **0.003 (0.003, 0.004); <0.001** | **0.004 (0.003, 0.005); <0.001** |
| **Smoking history** | | | | |
| • Yes | | | | -0.005 (-0.013, 0.003); 0.225 |
| • Don't know/refused | | | | -0.003 (-0.040, 0.034); 0.884 |
| • No | | | | *Reference* |

*Model I*, birth weight was adjusted for gestational age and sex. In *Model II*, apart from gestational age and sex, BMI and adult age was adjusted. Similarly, in *Model III*, apart from variables adjusted in *Model I* and *II*, the history of smoking was added.

*R$^2$ = 22.7%

*Abbreviations*: apoB/apoA1 ratio: Apolipoprotein B/Apolipoprotein A1 ratio; BMI: Body Mass Index; β-coeff: β-coefficient; CI: Confidence Interval

P-value<0.05 was considered statistically significant, with statistically significant associations in bold.

study concluded that cholesterol metabolism was most affected in individuals exposed to famine in early gestation [20].

Another plausible mechanism of early life influences on adult lipid markers could be the mediatory effect of adult body mass index (BMI) as documented in the study by Pehkonen *et al.* among young Finns [15]. The study showed that a 10% increase in birth weight was

**Table 5. Unadjusted and adjusted model of linear regression between *total cholesterol* (dependent variable) and independent variables.**

| Variables | Unadjusted Model | Model I | Model II | Model III * |
|---|---|---|---|---|
| | β-coeff. (95% CI); p-value | β-coeff. (95% CI); p-value | β-coeff. (95% CI); p-value | β-coeff. (95% CI); p-value |
| **Birth weight** (kg) | -0.026 (-0.058, 0.007); 0.122 | **-0.053 (-0.091, -0.014); 0.008** | -0.029 (-0.067, 0.009); 0.132 | -0.040 (-0.088, 0.008); 0.100 |
| **Gestational age** (weeks) | | 0.010 (-0.002, 0.021); 0.105 | -0.004 (-0.015, 0.008); 0.522 | -0.007 (-0.021, 0.007); 0.314 |
| **Sex** | | | | |
| Women | | **-0.096 (-0.131, -0.061); <0.001** | 0.002 (-0.033, 0.037); 0.903 | -0.008 (-0.053, 0.036); 0.722 |
| | | | | *Reference* |
| Men | | *Reference* | *Reference* | |
| **BMI** (kg/m$^2$) | | | **0.032 (0.026, 0.037); <0.001** | **0.033 (0.026, 0.039); <0.001** |
| **Age** (years) | | | **0.035 (0.032, 0.038); <0.001** | **0.038 (0.034, 0.041); <0.001** |
| **Smoking history** | | | | |
| • Yes | | | | 0.005 (-0.039, 0.048); 0.832 |
| • Don't know/refused | | | | 0.056 (-0.147, 0.258); 0.590 |
| • No | | | | *Reference* |

*Model I*, birth weight was adjusted for gestational age and sex. In *Model II*, apart from gestational age and sex, BMI and adult age was adjusted for. Similarly, in *Model III*, apart from variables adjusted in *Model I* and *II*, history of smoking was added.

*R$^2$ = 8.3%

*Abbreviations*: BMI: Body Mass Index; β-coeff: β-coefficient; CI: Confidence Interval

P-value<0.05 was considered statistically significant, with significant associations in bold.

**Table 6. Unadjusted and adjusted model of linear regression between log-transformed *triglyceride levels* (dependent variable) and independent variables.**

| Variables | Unadjusted Model | Model I | Model II | Model III * |
|---|---|---|---|---|
| | β-coeff. (95% CI); p-value | β-coeff. (95% CI); p-value | β-coeff. (95% CI); p-value | β-coeff. (95% CI); p-value |
| **Birth weight** (kg) | 0.003 (-0.031, 0.038); 0.844 | **-0.048 (-0.087, -0.009); 0.015** | **-0.057 (-0.094, -0.020); 0.003** | **-0.067 (-0.114, -0.021); 0.005** |
| **Gestational age** (weeks) | | 0.008 (-0.003, 0.019); 0.165 | 0.009 (-0.002, 0.020); 0.108 | 0.007 (-0.006, 0.021); 0.278 |
| **Sex** | | | | |
| Women | | **-0.324 (-0.360, -0.287);** | **-0.259 (-0.295, -0.223);** | **-0.262 (-0.306, -0.217);** |
| | | **<0.001** | **<0.001** | **<0.001** |
| Men | | *Reference* | *Reference* | *Reference* |
| **BMI** (kg/m$^2$) | | | **0.040 (0.035, 0.045); <0.001** | **0.040 (0.033, 0.046); <0.001** |
| **Age** (years) | | | 0.003 (-0.001, 0.007); 0.103 | 0.003 (-0.002, 0.007); 0.230 |
| **Smoking history** | | | | |
| • Yes | | | | 0.003 (-0.040, 0.046); 0.891 |
| • Don't know/refused | | | | -0.029 (-0.207, 0.149); 0.750 |
| • No | | | | *Reference* |

*Model I*, birth weight was adjusted for gestational age and sex. In *Model II*, apart from gestational age and sex, BMI and adult age was adjusted for. Similarly, in Model III, apart from variables adjusted in *Model I* and *II*, history of smoking was added.

*$R^2$ = 17.7%

*Abbreviations*: BMI: Body Mass Index; β-coeff: β-coefficient; CI: Confidence Interval

P-value<0.05 was considered statistically significant, with significant associations in bold.

associated with a 1.12% increase in adult BMI, which in turn was associated with a 1.1% increase in triglycerides [15].

It is crucial to explore apolipoproteins as they are potential risk markers for cardiovascular diseases [3]. A causal association between elevated lipoproteins and atherosclerotic disease manifestations has been documented in many genetic studies [22], for example, lipoprotein (a) [23]. Early prediction of CVD through elevated apolipoproteins can support preventive strategies and reduce the risk [5]. Besides, this confirms the already established evidence of the link between birth weight and CVD risk. Compared to those with no history of low birth weight, adults with low birth weight had an increment of 14% and 15% in the odds of CVD and coronary heart disease (CHD), respectively [24]. This meta-analysis by Mohsen *et al*. reported a U-shaped association between birth weight and CVD risk [24]. Furthermore, Liang *et al*. concluded that there is an interaction effect of low birth weight with adult obesity to increase the risk of CHD [25].

The DOHaD hypothesis also proposes other plausible mechanisms for the associations between birth weight and CVD risks, such as epigenetic modifications and hypersecretion of glucocorticoids during pregnancy [26]. Adverse intrauterine conditions may induce epigenetic modifications like DNA methylation and histone modifications, transmissible to the offspring. For example, the changed epigenetics of cardiomyocytes might induce ischemic diseases in adulthood [26].

## Limitations and strengths

The study results should be interpreted considering the following limitations. *Firstly*, the study was limited to the Swedish population, which makes it difficult to generalize the findings to other ethnicities or populations in, for example, developing countries. *Secondly*, the LifeGene cohort is a relatively young cohort by design [11] with a mean age of participants in these analyses is around 30 years. Results and some health indicators are therefore different from the

Swedish general population [27]. Hence, the influence on mean apolipoprotein levels may not have been fully evident as they occur predominantly in older age. Furthermore, the dataset lacked information on medication for hypercholesterolemia and about other cardiovascular risk conditions that influence CVD risk or apolipoproteins, such as Non-Alcoholic Fatty Liver Disease (NAFLD), and/or diabetes and hence, they were not adjusted for. However, according to the young age of the cohort these influences are supposed to be minor. In addition, blood samples for cholesterol and triglycerides were not collected in the fasting state, thereby adding bias to the results. Many adult behavioral factors such as lifestyle, physical activity, and dietary habits, but also genetic factors, are potential confounders and could influence the concentration of apolipoproteins but were not included in our study. *Lastly*, we had access to data on apolipoproteins apoA1 and apoB levels as well as standard variables such as total cholesterol and triglycerides, but not to other sub-types of lipoproteins, such as LDL and HDL cholesterol or lipoprotein (a). However, the study had many strengths. This is a population-based cohort with a large sample size of over 10,000 individuals and one of the few to cover both apoA1 and apoB levels in the population. The birth details of the study participants were obtained from the National Medical Birth Register with well-standardized data provided at birth from midwives [12] and thus not influenced by recall bias. Of special importance is to have high-quality data on gestational age as an important factor to adjust for based on register information, even if recall of birth weight could correspond to objective information on birth weight according to one meta-analysis [28].

## Conclusions

Lower birth weight was associated with higher apoB and lower apoA1 levels as well as higher apoB/apoA1 ratio in adult life, adjusted for gestational age and confounders. Such associations between birth weight and apolipoproteins could represent an early risk marker pattern of future CVD [29]. Further studies are warranted to better understand the independent or mediatory effect of apolipoproteins in increasing cardiovascular risk against a background of impaired fetal growth and lower birth weight.

## Supporting information

**S1 Data.**
(XLSX)

## Acknowledgments

We wish to thank Professor Nancy Pedersen, Karolinska Institute, the Principal Investigator of the LifeGene cohort, for her help with the data set, and data manager Anders Dahlin, Malmö, for technical support.

## Author Contributions

**Conceptualization:** Shantanu Sharma, Peter M. Nilsson.

**Formal analysis:** Shantanu Sharma.

**Project administration:** Peter M. Nilsson.

**Resources:** Peter M. Nilsson.

**Supervision:** Louise Bennet, Agne Laucyte-Cibulskiene, Anders Christensson, Peter M. Nilsson.

**Visualization:** Peter M. Nilsson.

**Writing – original draft:** Shantanu Sharma.

**Writing – review & editing:** Louise Bennet, Agne Laucyte-Cibulskiene, Anders Christensson, Peter M. Nilsson.

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
