## [Decision Letter · Decision Letter 0]

24 Oct 2023

PONE-D-23-17201Associations between birth variables and adult apolipoproteins: The LifeGene CohortPLOS ONE

Dear Dr. Sharma,

Thank you for submitting your manuscript to PLOS ONE. After careful consideration, we feel that it has merit but does not fully meet PLOS ONE’s publication criteria as it currently stands. Therefore, we invite you to submit a revised version of the manuscript that addresses the points raised during the review process.

We look forward to receiving your revised manuscript.

Kind regards,

Enoch Odame Anto

Academic Editor

PLOS ONE

Journal Requirements:

"The LifeGene Study has been funded by the Torsten and Ragnar Söderbergs Foundation, AFA Insurance, Karolinska Institute, the Stockholm County Council, and the Swedish Research Council."

"We wish to thank Professor Nancy Pedersen, Karolinska Institute, the Principal Investigator of the LifeGene cohort, for her help with the data set, and data manager Anders Dahlin, Malmö, for technical support.

The LifeGene cohort was supported by start-up funding received from Karolinska Institute, the Stockholm County Council, and the Swedish Research Council. Funding has also been obtained from the Torsten and Ragnar Söderbergs Foundation and AFA Försäkringar."

"The LifeGene Study has been funded by the Torsten and Ragnar Söderbergs Foundation, AFA Insurance, Karolinska Institute, the Stockholm County Council, and the Swedish Research Council."

7. Your ethics statement should only appear in the Methods section of your manuscript. If your ethics statement is written in any section besides the Methods, please move it to the Methods section and delete it from any other section. Please ensure that your ethics statement is included in your manuscript, as the ethics statement entered into the online submission form will not be published alongside your manuscript. 

Additional Editor Comments:

Reviewer 1,

Review

Title: Associations between birth variables and adult apolipoproteins: The LifeGene Cohort

Abstract

- Methods: the sentence in lines 70 to 72 should be the last sentence of your methods section. You may want to mention the things you did first and the last ones last.

Results

- Line 77 should go to the methods section.

- Who was the majority between males and females? Include that information with frequency (%).

- Show the results for triglycerides. This sentence sounds like a conclusion “No independent association was found with total cholesterol, but with triglyceride levels”

Conclusion

- Is there a clinical or public health recommendation that authors can give based on their findings? It would be nice to see it.

Keywords- Line 88, how is epidemiology a keyword when there is no mention of it in the abstract?

Introduction

- Line 99 delete etc. and be specific. You probably have to add dyslipidemia, hypertension and abdominal obesity which are components of metabolic syndrome.

Materials and Methods

- Line 120, should be “Study participants”

- Line 125, sampling or sample?

- Line 127, what was provided electronically?

- Lines 128 to 129, apart from the in-person testing, what was the other way of testing? Please include some details.

- Figure 1, last box, what do you mean by “staring”? It would be nice also to see a box with the number of participants on which the analysis was based for this study.

- The sentence beginning line 148 and ending line 150 should go to the section on ethical considerations. If you don’t have one, please have one and take all the ethical issues there.

- Study variables: authors should state the primary outcome/s (dependent variable) and secondary outcome/s

- Line 156, “Have you smoked more than 100 cigarettes in your entire life?” cite where you obtained this standard. Does this mean that if one smokes more than 100 cigarettes then they have a heightened risk of CVD?

- Lines 157 to 158, I don’t think this is part of the “study variable” but a study procedure “The non-fasting blood samples were drawn at LifeGene test centers”, revise accordingly. I encourage the authors to have a subsection on “Study procedures” separate from the “study variables” so that lines 157 to 163 can go there.

- Authors should have a section for “definitions” e.g the smoking of 100 cigarettes, calculation of BMI and apoB/apoA1 ratio.

- Lines 164 to 166, delete because it has already been stated above (section on study sample)

Results

- Line 184, how many were males and females? Only state the majority.

- I don’t understand why Table 1 was based on sex (male and female), does it mean, the authors were also looking at the sex differences in the outcomes? If not please delete.

- This section can be divided as follows:

o 1. “Characteristics of the study participants”, where you describe the distribution of different study variables, NO need to show the table.

o 2. Table 1 association between ApoA1 and independent variables

o 3. Table 2 do the same for ApoB

o 4. Table 3 do the same for ApoA1/ApoB ratio

o 5. Table 4 do the same for triglycerides

o 6. Table 5 do the same for total cholesterol.

- NOTE: each table should have its own subheading and narrations. All the narrations should be above the table. Narrate the significant findings first.

- All significant findings in the models should be narrated/interpreted.

- Since total cholesterol is part of the outcome, the results should be included in the main text NOT as supplementary.

Discussion

- Line 251, delete the p-value. Do the same for line 257

- The section is well-written

Conclusion.

- The section is well-written

Reviewer 2

The manuscript by Sharma et al “’Associations between birth variables and adult apolipoproteins: The LifeGene Cohort” is interesting and presents original findings that add new information to literature. The sample size is large enough to make valid inferences. The paper can benefit from a few suggestions to further improve it.

Minor

1. This study is a cross-section study using participants from another prospective study (The LifeGene Study). This needs to be clear in the methods of both the abstract and main text

2. Am not sure if the usage of “adult” before “apolipoprotein” is appropriate in the title and abstract. Does adult refer to the type of apolipoproteins (not sure if we do refer to them as such) or the age category of the participant being studied? Consider editing the title/abstract to avoid the confusion. Also, “variables” can be substituted with “characteristics” or another term.

3. It appears birth weight and gestation age are the only birth characteristics referred to and mostly birth weight appears to be central to the study. I would suggest for the title to be more reflective of the main findings. However, this is not major and feel free to ignore

4. The primary outcome variable needs to be specified for the manuscript to be more focused to avoid a fishing expedition type of aim: is it apoA1 or apoB or their ratio? In my opinion, apoB/ApoA1 ratio should be the primary out as it is a better risk marker than individual apolipoproteins under discussion.

5. Abstract lines 82-83. It’s no clear which outcome was not associated with cholesterol/was associated with triglycerides

6. Lines 106-107. The authors themselves have indicated that Multiple factors, including lifestyle, dietary habits, and genetic makeup, influence the serum concentration of apolipoproteins and yet they have not clearly written this in their limitation section for related factors that they did not include in their study but with potential to confound the results

7. Inclusion criteria should be more specific i.e what was the age of inclusion? Adults aged 18 years and above or even children where included? See lines 121 vs 126

8. The flow of the Methods section needs some editing. Please follow reporting guidelines https://www.equator-network.org/ . The strobe checklist is used but reporting ad writing format especially in methods does not clearly indicate the subheadings for proper flow

9. In the results section it’s better to indicate the “n” for each variable and only for variables that had an “n” lower than the actual sample size. In Table 1, the meaning of N (sample)* is not clear

10. What criteria was used to arrive at 1 100 cigarettes? Why not 50 or 200 etc? See table 1

11. Table 2 tittle does not reflect what is in the table. Further the words in the cells /rows are not aligned properly. Consider using a row for each subcategory vs using one row for each variable combined with subcategories

Reviewers' comments:

Reviewer's Responses to Questions

**Comments to the Author**

1. Is the manuscript technically sound, and do the data support the conclusions?

Reviewer #1: Yes

Reviewer #2: Yes

2. Has the statistical analysis been performed appropriately and rigorously? 

Reviewer #1: Yes

Reviewer #2: Yes

3. Have the authors made all data underlying the findings in their manuscript fully available?

Reviewer #1: No

Reviewer #2: Yes

4. Is the manuscript presented in an intelligible fashion and written in standard English?

Reviewer #1: Yes

Reviewer #2: Yes

5. Review Comments to the Author

Reviewer #1: Review

Title: Associations between birth variables and adult apolipoproteins: The LifeGene Cohort

Abstract

- Methods: the sentence in lines 70 to 72 should be the last sentence of your methods section. You may want to mention the things you did first and the last ones last.

Results

- Line 77 should go to the methods section.

- Who was the majority between males and females? Include that information with frequency (%).

- Show the results for triglycerides. This sentence sounds like a conclusion “No independent association was found with total cholesterol, but with triglyceride levels”

Conclusion

- Is there a clinical or public health recommendation that authors can give based on their findings? It would be nice to see it.

Keywords- Line 88, how is epidemiology a keyword when there is no mention of it in the abstract?

Introduction

- Line 99 delete etc. and be specific. You probably have to add dyslipidemia, hypertension and abdominal obesity which are components of metabolic syndrome.

Materials and Methods

- Line 120, should be “Study participants”

- Line 125, sampling or sample?

- Line 127, what was provided electronically?

- Lines 128 to 129, apart from the in-person testing, what was the other way of testing? Please include some details.

- Figure 1, last box, what do you mean by “staring”? It would be nice also to see a box with the number of participants on which the analysis was based for this study.

- The sentence beginning line 148 and ending line 150 should go to the section on ethical considerations. If you don’t have one, please have one and take all the ethical issues there.

- Study variables: authors should state the primary outcome/s (dependent variable) and secondary outcome/s

- Line 156, “Have you smoked more than 100 cigarettes in your entire life?” cite where you obtained this standard. Does this mean that if one smokes more than 100 cigarettes then they have a heightened risk of CVD?

- Lines 157 to 158, I don’t think this is part of the “study variable” but a study procedure “The non-fasting blood samples were drawn at LifeGene test centers”, revise accordingly. I encourage the authors to have a subsection on “Study procedures” separate from the “study variables” so that lines 157 to 163 can go there.

- Authors should have a section for “definitions” e.g the smoking of 100 cigarettes, calculation of BMI and apoB/apoA1 ratio.

- Lines 164 to 166, delete because it has already been stated above (section on study sample)

Results

- Line 184, how many were males and females? Only state the majority.

- I don’t understand why Table 1 was based on sex (male and female), does it mean, the authors were also looking at the sex differences in the outcomes? If not please delete.

- This section can be divided as follows:

o 1. “Characteristics of the study participants”, where you describe the distribution of different study variables, NO need to show the table.

o 2. Table 1 association between ApoA1 and independent variables

o 3. Table 2 do the same for ApoB

o 4. Table 3 do the same for ApoA1/ApoB ratio

o 5. Table 4 do the same for triglycerides

o 6. Table 5 do the same for total cholesterol.

- NOTE: each table should have its own subheading and narrations. All the narrations should be above the table. Narrate the significant findings first.

- All significant findings in the models should be narrated/interpreted.

- Since total cholesterol is part of the outcome, the results should be included in the main text NOT as supplementary.

Discussion

- Line 251, delete the p-value. Do the same for line 257

- The section is well-written

Conclusion.

- The section is well-written

Reviewer #2: The manuscript by Sharma et al “’Associations between birth variables and adult apolipoproteins: The LifeGene Cohort” is interesting and presents original findings that add new information to literature. The sample size is large enough to make valid inferences. The paper can benefit from a few suggestions to further improve it.

Minor

1. This study is a cross-section study using participants from another prospective study (The LifeGene Study). This needs to be clear in the methods of both the abstract and main text

2. Am not sure if the usage of “adult” before “apolipoprotein” is appropriate in the title and abstract. Does adult refer to the type of apolipoproteins (not sure if we do refer to them as such) or the age category of the participant being studied? Consider editing the title/abstract to avoid the confusion. Also, “variables” can be substituted with “characteristics” or another term.

3. It appears birth weight and gestation age are the only birth characteristics referred to and mostly birth weight appears to be central to the study. I would suggest for the title to be more reflective of the main findings. However, this is not major and feel free to ignore

4. The primary outcome variable needs to be specified for the manuscript to be more focused to avoid a fishing expedition type of aim: is it apoA1 or apoB or their ratio? In my opinion, apoB/ApoA1 ratio should be the primary out as it is a better risk marker than individual apolipoproteins under discussion.

5. Abstract lines 82-83. It’s no clear which outcome was not associated with cholesterol/was associated with triglycerides

6. Lines 106-107. The authors themselves have indicated that Multiple factors, including lifestyle, dietary habits, and genetic makeup, influence the serum concentration of apolipoproteins and yet they have not clearly written this in their limitation section for related factors that they did not include in their study but with potential to confound the results

7. Inclusion criteria should be more specific i.e what was the age of inclusion? Adults aged 18 years and above or even children where included? See lines 121 vs 126

8. The flow of the Methods section needs some editing. Please follow reporting guidelines https://www.equator-network.org/ . The strobe checklist is used but reporting ad writing format especially in methods does not clearly indicate the subheadings for proper flow

9. In the results section it’s better to indicate the “n” for each variable and only for variables that had an “n” lower than the actual sample size. In Table 1, the meaning of N (sample)* is not clear

10. What criteria was used to arrive at 1 100 cigarettes? Why not 50 or 200 etc? See table 1

11. Table 2 tittle does not reflect what is in the table. Further the words in the cells /rows are not aligned properly. Consider using a row for each subcategory vs using one row for each variable combined with subcategories

6. PLOS authors have the option to publish the peer review history of their article (what does this mean?). If published, this will include your full peer review and any attached files.

Reviewer #1: No

Reviewer #2: No

---

## [Author Response · Author response to Decision Letter 0]

30 Dec 2023

Query 1: Thank you for stating the following financial disclosure: 

"The LifeGene Study has been funded by the Torsten and Ragnar Söderbergs Foundation, AFA Insurance, Karolinska Institute, the Stockholm County Council, and the Swedish Research Council." Please state what role the funders took in the study. If the funders had no role, please state: ""The funders had no role in study design, data collection and analysis, decision to publish, or preparation of the manuscript."" 

Response: We added the following sentence: "The funders had no role in study design, data collection and analysis, decision to publish, or preparation of the manuscript.” 

Query 2: Thank you for stating the following in the Acknowledgments Section of your manuscript: "We wish to thank Professor Nancy Pedersen, Karolinska Institute, the Principal Investigator of the LifeGene cohort, for her help with the data set, and data manager Anders Dahlin, Malmö, for technical support.

The LifeGene cohort was supported by start-up funding received from Karolinska Institute, the Stockholm County Council, and the Swedish Research Council. Funding has also been obtained from the Torsten and Ragnar Söderbergs Foundation and AFA Försäkringar."

"The LifeGene Study has been funded by the Torsten and Ragnar Söderbergs Foundation, AFA Insurance, Karolinska Institute, the Stockholm County Council, and the Swedish Research Council." Please include your amended statements within your cover letter; we will change the online submission form on your behalf.

Response: OK, I have now removed the funding section from the acknowledgement section.

Query 3: In your Data Availability statement, you have not specified where the minimal data set underlying the results described in your manuscript can be found. PLOS defines a study's minimal data set as the underlying data used to reach the conclusions drawn in the manuscript and any additional data required to replicate the reported study findings in their entirety. All PLOS journals require that the minimal data set be made fully available. For more information about our data policy, please see http://journals.plos.org/plosone/s/data-availability.

Upon re-submitting your revised manuscript, please upload your study’s minimal underlying data set as either Supporting Information files or to a stable, public repository and include the relevant URLs, DOIs, or accession numbers within your revised cover letter. For a list of acceptable repositories, please see http://journals.plos.org/plosone/s/data-availability#loc-recommended-repositories. Any potentially identifying patient information must be fully anonymized. Important: If there are ethical or legal restrictions to sharing your data publicly, please explain these restrictions in detail. Please see our guidelines for more information on what we consider unacceptable restrictions to publicly sharing data: http://journals.plos.org/plosone/s/data-availability#loc-unacceptable-data-access-restrictions. Note that it is not acceptable for the authors to be the sole named individuals responsible for ensuring data access. We will update your Data Availability statement to reflect the information you provide in your cover letter.

Response: Please find the enclosed file with supporting anonymized minimal dataset. This is according to the requirements of PlosOne that we want to comply with. We hope that this procedure will be acceptable for you. If not, please inform us accordingly.

Query 4: We note that you have stated that you will provide repository information for your data at acceptance. Should your manuscript be accepted for publication, we will hold it until you provide the relevant accession numbers or DOIs necessary to access your data. If you wish to make changes to your Data Availability statement, please describe these changes in your cover letter and we will update your Data Availability statement to reflect the information you provide.

Response: We have modified the data availability statement: ”The anonymized minimal dataset is available. For details about the LifeGene cohort and instructions on how to apply for data, see link: www.lifegene.se .

Query 5: Your ethics statement should only appear in the Methods section of your manuscript. If your ethics statement is written in any section besides the Methods, please move it to the Methods section and delete it from any other section. Please ensure that your ethics statement is included in your manuscript, as the ethics statement entered into the online submission form will not be published alongside your manuscript. 

Response: OK, it has now been shifted to the Methodology section.

Additional Editor Comments:

Reviewer 1:

Review

Title: Associations between birth variables and adult apolipoproteins: The LifeGene Cohort

Abstract

Query: - Methods: the sentence in lines 70 to 72 should be the last sentence of your methods section. You may want to mention the things you did first and the last ones last.

Response: Done

Results

Query: - Line 77 should go to the methods section.

Response: Done

Query: - Who was the majority between males and females? Include that information with frequency (%).

Responses: We have now added: “Out of 10,093, nearly 42.5% were men (n=4292) and 57.5% were women (n=5801).”

Query: - Show the results for triglycerides. This sentence sounds like a conclusion “No independent association was found with total cholesterol, but with triglyceride levels”

Response: Added “(ẞ-coefficient (95% Confidence Interval) -0.067 (-0.114, -0.021); p-value 0.005).”

Conclusion

Query: - Is there a clinical or public health recommendation that authors can give based on their findings? It would be nice to see it.

Response: Added “The study highlights the need of preconception care and pregnancy interventions that aim at improving maternal and child outcomes with long-term health impacts for prevention of cardiovascular disease by influencing lipid levels.”

Query: Keywords- Line 88, how is epidemiology a keyword when there is no mention of it in the abstract?

Response: Removed.

Introduction

Query: - Line 99 delete etc. and be specific. You probably have to add dyslipidemia, hypertension and abdominal obesity which are components of metabolic syndrome.

Response: Now added.

Materials and Methods

Query- Line 120, should be “Study participants”

Response: Done

Query- Line 125, sampling or sample?

Response: Done 

Query- Line 127, what was provided electronically?

Response: Now the text has been revised.

Query- Lines 128 to 129, apart from the in-person testing, what was the other way of testing? 

Response: Now the text has been revised.

Please include some details.

Query- Figure 1, last box, what do you mean by “staring”? It would be nice also to see a box with the number of participants on which the analysis was based for this study.

Response: Now corrected as “Starting”. The last box states 10,093, that is the final number (sample size) of participants on whom our analysis was based.

Query- The sentence beginning line 148 and ending line 150 should go to the section on ethical considerations. If you don’t have one, please have one and take all the ethical issues there.

Response: We have now added a separate section for “Ethics” of the study.

Query- Study variables: authors should state the primary outcome/s (dependent variable) and secondary outcome/s

Response: Primary outcomes (dependent variables) are already mentioned. There are no secondary outcomes. The word “primary” has been added.

Query- Line 156, “Have you smoked more than 100 cigarettes in your entire life?” cite where you obtained this standard. Does this mean that if one smokes more than 100 cigarettes then they have a heightened risk of CVD?

Response: This question is used in various surveys in the past like the ones conducted by NIH, US, cited below and thus well-known. 

References:

1. Bondy SJ, Victor JC, Diemert LM. Origin and use of the 100 cigarette criterion in tobacco surveys. Tobacco Control. 2009;18(4):317–323

2. Klemperer EM, Hughes JR, Callas PW, West JC, Villanti AC. Tobacco and Nicotine Use Among US Adult "Never Smokers" in Wave 4 (2016-2018) of the Population Assessment of Tobacco and Health Study. Nicotine Tobacco Res. 2021 Jun 8;23(7):1199-1207.

Query- Lines 157 to 158, I don’t think this is part of the “study variable” but a study procedure “The non-fasting blood samples were drawn at LifeGene test centers”, revise accordingly. I encourage the authors to have a subsection on “Study procedures” separate from the “study variables” so that lines 157 to 163 can go there.

Response: Done

Query- Authors should have a section for “definitions” e.g the smoking of 100 cigarettes, calculation of BMI and apoB/apoA1 ratio.

Response: We have now added definitions for BMI and the apoB/apoA1 ratio

Query- Lines 164 to 166, delete because it has already been stated above (section on study sample)

Response: No repetition, as two different laboratories were involved for the analyses. This has now been clarified.

Results

Query- Line 184, how many were males and females? Only state the majority.

Response: The majority (57.5%) of the participants were women (n=5801).

Query- I don’t understand why Table 1 was based on sex (male and female), does it mean, the authors were also looking at the sex differences in the outcomes? If not please delete.

Response: We wanted to show the distribution of variables among men and women

Query- This section can be divided as follows:

o 1. “Characteristics of the study participants”, where you describe the distribution of different study variables, NO need to show the table.

Response: It is customary to show characteristics of study participants in a separate Table why we prefer to keep Table 1 for clarity. Most journals ask authors for this.

o 2. Table 1 association between ApoA1 and independent variables

o 3. Table 2 do the same for ApoB

o 4. Table 3 do the same for ApoA1/ApoB ratio

o 5. Table 4 do the same for triglycerides

o 6. Table 5 do the same for total cholesterol.

Query- NOTE: each table should have its own subheading and narrations. All the narrations should be above the table. Narrate the significant findings first.

Response: We wanted to avoid too much text because of word limitations, and to avoid duplication of data presentation both in the text and in the respective Table.

Query- All significant findings in the models should be narrated/interpreted.

Response: We wanted to avoid too much text because of word limitations, and to avoid duplication of data presentation both in the text and in several Tables. This is an aspect that we ask the Editor to guide us about if a change is asked for.

Query- Since total cholesterol is part of the outcome, the results should be included in the main text NOT as supplementary.

Response: So, I have now shifted the Supplementary Tables into main Tables (Table 5 and Table 6).

Discussion

Query- Line 251, delete the p-value. Do the same for line 257

Response: Done

- The section is well-written

Conclusion.

- The section is well-written

Reviewer 2

The manuscript by Sharma et al “’Associations between birth variables and adult apolipoproteins: The LifeGene Cohort” is interesting and presents original findings that add new information to literature. The sample size is large enough to make valid inferences. The paper can benefit from a few suggestions to further improve it.

Minor

Query-This study is a cross-section study using participants from another prospective study (The LifeGene Study). This needs to be clear in the methods of both the abstract and main text

Response: We have now clarified that our study sample represents a sub-study of the original LifeGene study cohort (n= 28 000).

Query-Am not sure if the usage of “adult” before “apolipoprotein” is appropriate in the title and abstract. Does adult refer to the type of apolipoproteins (not sure if we do refer to them as such) or the age category of the participant being studied? Consider editing the title/abstract to avoid the confusion. Also, “variables” can be substituted with “characteristics” or another term.

Response: The “adult” term was used because we selected individuals 18 years or older at screening and removed all individuals below 18 years of age. Other studies deal with apoliprotein patterns in children and adolescents (Sipola-Leppänen M, et al. Cardiovascular risk factors in adolescents born preterm. Pediatrics. 2014 Oct;134(4):e1072-81; Taageby Nielsen S, et al. Significance of lipids, lipoproteins, and apolipoproteins during the first 14-16 months of life. Eur Heart J. 2023 Nov 7;44(42):4408-4418), why this topic is already covered. We have changed to “characteristics”. 

Query-It appears birth weight and gestation age are the only birth characteristics referred to and mostly birth weight appears to be central to the study. I would suggest for the title to be more reflective of the main findings. However, this is not major and feel free to ignore

Response: We prefer a shorter title like the one at present, but have now changed “birth variables” to “birth weight”, as recommended by the reviewer.

Query-The primary outcome variable needs to be specified for the manuscript to be more focused to avoid a fishing expedition type of aim: is it apoA1 or apoB or their ratio? In my opinion, apoB/ApoA1 ratio should be the primary out as it is a better risk marker than individual apolipoproteins under discussion.

Response: Every ratio depends on the variables included. This is why most journals prefer to list the individual variables first and the corresponding ratio following that.

Query-Abstract lines 82-83. It’s no clear which outcome was not associated with cholesterol/was associated with triglycerides

Response: Clarification has been carried out.

Query-Lines 106-107. The authors themselves have indicated that Multiple factors, including lifestyle, dietary habits, and genetic makeup, influence the serum concentration of apolipoproteins and yet they have not clearly written this in their limitation section for related factors that they did not include in their study but with potential to confound the results

Response: We have added “Many adult behavioural factors such as lifestyle, physical activity, and dietary habits, but also genetic factors are potential confounders and could influence the concentration of apolipoproteins, but were not included in our study.”

Query-Inclusion criteria should be more specific i.e. what was the age of inclusion? Adults aged 18 years and above or even children where included? See lines 121 vs 126

Responses: No children below 18 years were included, by design. See reply to Reviewer 1 above.

Query-The flow of the Methods section needs some editing. Please follow reporting guidelines https://www.equator-network.org/ . The strobe checklist is used but reporting ad writing format especially in methods does not clearly indicate the subheadings for proper flow

Response: All corrections have been accomplished.

Query-In the results section it’s better to indicate the “n” for each variable and only for variables that had an “n” lower than the actual sample size. In Table 1, the meaning of N (sample)* is not clear

Response: We have specified the total sample size of men and women in Table 1. Then, it’s the mean/median with SD/IQR that has been mentioned except for the last row that had N (which is the frequency), which has been clarified in the footnote.

Query-What criteria was used to arrive at 1 100 cigarettes? Why not 50 or 200 etc? See table 1

Responses: This question was previously used in various surveys like the one conducted by NIH, US, as cited below. 

References:

1. Bondy SJ, Victor JC, Diemert LM. Origin and use of the 100 cigarette criterion in tobacco surveys. Tob Control. 2009;18(4):317–323

2. Klemperer EM, Hughes JR, Callas PW, West JC, Villanti AC. Tobacco and Nicotine Use Among US Adult "Never Smokers" in Wave 4 (2016-2018) of the Population Assessment of Tobacco and Health Study. Nicotine Tob Res. 2021 Jun 8;23(7):1199-1207.

Query-Table 2 tittle does not reflect what is in the table. Further the words in the cells /rows are not aligned properly. Consider using a row for each subcategory vs using one row for each variable combined with subcategories

Response: The titles of Tables 2 and 3 have been revised for clarity. Because of the font size, the rows were not properly arranged. I have now reduced the font size and the rows look better now. Only in the last row, there are subcategories, which have been collapsed into one cell to avoid confusion to the reader.

---

## [Decision Letter · Decision Letter 1]

16 Feb 2024

Associations between birth weight and adult apolipoproteins: The LifeGene Cohort

PONE-D-23-17201R1

Dear Dr. Sharma,

We’re pleased to inform you that your manuscript has been judged scientifically suitable for publication and will be formally accepted for publication once it meets all outstanding technical requirements.

Kind regards,

Enoch Odame Anto

Academic Editor

PLOS ONE

---

## [Editor Report · Acceptance letter]

22 Feb 2024

PONE-D-23-17201R1 

PLOS ONE

Dear Dr. Sharma, 

I'm pleased to inform you that your manuscript has been deemed suitable for publication in PLOS ONE. Congratulations! Your manuscript is now being handed over to our production team.

Kind regards, 

on behalf of

Dr. Enoch Odame Anto 

Academic Editor

PLOS ONE